# HIV-1 Latency and Latency Reversal: Does Subtype Matter?

**DOI:** 10.3390/v11121104

**Published:** 2019-11-28

**Authors:** Indra Sarabia, Alberto Bosque

**Affiliations:** Department of Microbiology, Immunology and Tropical Medicine, The George Washington University, Washington, DC 20052, USA; sarabia@email.gwu.edu

**Keywords:** HIV-1, HIV-1 latency, shock and kill, clade, subtype

## Abstract

Cells that are latently infected with HIV-1 preclude an HIV-1 cure, as antiretroviral therapy does not target this latent population. HIV-1 is highly genetically diverse, with over 10 subtypes and numerous recombinant forms circulating worldwide. In spite of this vast diversity, much of our understanding of latency and latency reversal is largely based on subtype B viruses. As such, most of the development of cure strategies targeting HIV-1 are solely based on subtype B. It is currently assumed that subtype does not influence the establishment or reactivation of latent viruses. However, this has not been conclusively proven one way or the other. A better understanding of the factors that influence HIV-1 latency in all viral subtypes will help develop therapeutic strategies that can be applied worldwide. Here, we review the latest literature on subtype-specific factors that affect viral replication, pathogenesis, and, most importantly, latency and its reversal.

## 1. Introduction

Human Immunodeficiency Virus (HIV-1) remains one of the highest causes of morbidity and mortality worldwide. Antiretroviral therapy (ART) has significantly improved the lives of those living with HIV-1, but a daily commitment to these medications is critical to managing the disease. If ART is interrupted, the virus can resurge, and the disease progresses to Acquired Immunodeficiency Syndrome (AIDS). This is because HIV-1 is able to establish latency [1,2,3,4]. Latently infected cells harbor an integrated but transcriptionally silent HIV-1 provirus that can be reactivated [5]. ART targets different steps of the viral life cycle of HIV-1, such as reverse transcription, viral entry, integration, and viral budding, but cannot target these latently infected cells [1,2,3,4,5,6,7]. Thus, these latently infected cells are a barrier to an HIV-1 cure. Resting memory CD4 T cells are thought to be one of the main latent reservoirs, though other cell types contribute to this reservoir [8,9,10,11,12,13]. Several strategies are being pursued towards eliminating this latent reservoir [14]. “Shock-and-kill” is a cure strategy currently in clinical trials (reviewed in [15,16]). This strategy involves revealing latently infected cells to the immune system by using latency reversing agents (LRAs). LRAs can promote viral protein expression through several distinct mechanisms, such as relieving repressive epigenetic modifications or supplying host transcription factors and other cellular factors necessary for viral gene expression, thus producing viral antigens (“Shock”) [17]. Reactivated cells are then revealed and are now sensitive to the cytotoxic arm of the immune system (“Kill”).

Significant progress has been made in developing novel LRAs and understanding the mechanisms underlying HIV-1 latency reversal, but these findings have been mostly conducted with subtype B viruses. The genetic diversity of HIV-1 is staggering, with over 10 subtypes worldwide and ever-increasing recombinant strains [18]. However, it is incompletely understood whether subtype has a biological impact on latency establishment and latency reversal. This complexity can become a challenge when developing a curative strategy to be implemented worldwide. For example, the vast genetic diversity of HIV-1 has long been a challenge to the development of a universal HIV-1 vaccine. The envelope protein, the main epitope utilized for vaccine development, can vary greatly from subtype to subtype, affecting vaccine efficacy [19]. Thus, the development of an HIV-1 vaccine that can provide cross-subtype protection is vital. This same reasoning should be applied to HIV-1 cure strategies, where the methods developed should elicit the most latency reversal across subtypes, not just subtype B. Currently, it is not well characterized as to whether subtype has an impact on latency or latency reversal. As such, understanding whether subtype may affect latency should be addressed early on in the development of curative strategies to be implemented worldwide.

Here, we provide a comprehensive literature review specifically focused on differences in viral pathogenesis, co-receptor usage, viral protein function, and viral long terminal repeat (LTR) sequence among subtypes and how this may influence the establishment and reactivation of the latent reservoir.

## 2. Role of Subtype in HIV-1 Pathogenesis

Within HIV-1 group M viruses, there are at least 10 subtypes and numerous circulating recombinant forms (CRFs) and unique recombinant forms (URFs). They are classified as A, B, C, D, F, G, H, J, K, and major CRFs such as AE and AG (CRF01_AE, CRF02_AG). Subtypes can be further divided into sub-subtypes, which are denoted with numbers, e.g., subtype A1 [20]. Subtype AE was previously described as subtype E (and is referred as such in older publications), but it has been shown to be a recombinant form of subtype A and was reclassified as subtype AE [20]. The distribution of subtype worldwide is not uniform, as shown in Figure 1. Geographically, subtype C has been shown to mainly circulate in the sub-Saharan African continent. Subtype G circulates in Western and Central African countries, where some CRFs also exist, such as AG. Western Europe is mostly subtype B, with some small populations of subtypes A, C, G, and AG. Eastern Europe is mostly subtype A, with a population of subtype B and smaller fractions of subtypes C, F, and AB. Northern American countries, Latin America, and the Caribbean are largely subtype B with some populations of subtypes C and F. South and Southeastern Asia is largely subtype AE, while Eastern Asia is comprised of subtypes B, AE, and subtype BC recombinants [18,21,22,23]. The locations of these various subtypes are not fixed in place, as there are reports showing an increased diversity of subtypes in traditionally subtype B-dominant regions [24,25,26]. Recent work by Yamaguchi et al. have helped establish the presence of a new subtype, L, through next-generation sequencing [27]. Using country-specific HIV-1 molecular epidemiology data from 2000 to 2007, Hemelaar et al. generated global estimates for the prevalence of the distinct subtypes. Subtype C comprises approximately 48% of infections, while subtype A comprises 12%, subtype B 11%, subtype G 5%, and subtype D 2% of worldwide infections. Subtypes F, H, J, and K are a relative global minority and comprise 0.45%, 0.26%, 0.12%, and 0.01%, respectively. Recombinant CRF and URF subtypes comprise a large portion of global infections, about 20%. The major CRFs, AE and AG, comprise approximately 5% and 8%, respectively [18,21,22,23].

Several studies have examined the association between viral subtype and disease progression. The results of these studies which examined pathogenesis in vivo are explained in detail below and are summarized in Table 1. Earlier studies found no difference in disease progression among viral subtypes [28,29,30], though most recent studies did find a significant difference [31,32,33,34,35,36,37]. These opposing conclusions may be due to differences in sample size, host or environmental factors, disease stage of participants, or subtype identification methods. Additionally, making direct comparisons between subtypes can be challenging, as any underlying differences in host or environmental factors can confound the results of the study [18]. Studies that examine pathogenesis where the subtypes co-circulate within the same population are the most appropriate, as it presumably eliminates some of these confounding variables. For example, in a prospective study involving Senegalese female sex workers, women who were infected with non-subtype A virus had a greater risk of developing AIDS than subtype-A-infected women in the same cohort [38]. Similarly, individuals infected with subtype D had a faster CD4 decline when compared to subtype A-infected individuals in a seroincident cohort in Uganda [31]. Baeten et al. also observed an association of subtype and faster CD4 decline in a cohort of Kenyan commercial sex workers, who were followed from the time of HIV-1 acquisition [32]. In this study, individuals infected with subtype D virus were also found to be at a higher risk of death compared to subtype A, and this risk persisted after correction for plasma HIV-1 viral loads. Similarly, Ssemwanga et al. found that subtype D-infected individuals had faster CD4 decline when compared to subtype A-infected individuals in rural Uganda [34]. In another study, subtype D infection was associated with the fastest CD4 decline, compared to subtype A or C, in a Tanzanian cohort of pregnant mothers [35]. Furthermore, subtype D infection was associated with the greatest CD4 cell decline in a cohort of women in Sub-Saharan Africa who were either subtype A, C, or D [36]. Kaleebu et al. found similar results within a cohort of >1000 subtype A or subtype D individuals, where subtype D individuals had faster disease progression compared to subtype A [33]. Finally, Amornkul et al. assessed disease progression in a cohort of subtypes A, C, and D individuals and found similar results where non-subtype A individuals had faster CD4 decline compared to subtype A individuals [37]. Taken together, the overall consensus is that there are differences in disease progression associated with viral subtype. This has obvious implications for the treatment of HIV-1, but it is less clear whether this difference in pathogenesis will impact latency establishment or latency-reversal.

## 3. HIV-1 Coreceptor Usage and Tropism Switch

As untreated HIV-1 infection progresses, the virus can switch from CCR5 to CXCR4 usage [39,40,41]. This switch to CXCR4 is correlated with disease progression [40], which is typical of subtype B viruses and can emerge late in disease in other subtypes as well [18,42]. The HIV-1 envelope, being the only protein that is exposed, is a target for antibody and cell-mediated immune responses and is obviously indispensable for entry into host cells (reviewed in [43]). As such, the sequence diversity within the viral *env* gene has been characterized extensively, mostly with subtype B and C. Between subtypes, the sequence identity of the *env* gene can vary by as much as 35% (for a review on Env diversity, see [44]). The sequence of the third variable loop (V3 loop) of the viral glycoprotein gp120 is critical for infection and is a determinant of coreceptor usage [45,46,47]. Interestingly, not all HIV-1 subtypes uniformly switch coreceptor usage, even in late stages of the disease. Subtype C and subtype A rarely undergo this switch; subtype C viruses favor CCR5 more extremely throughout infection than subtype A [20,48,49]. The V3 loop sequence length, amino acid charge, glycosylation site presence, and amino acid variations affect the development of CXCR4 usage [50]. Subtype C exhibits less sequence variation in the V3 loop compared to subtype B. Subtype A has been reported to be highly similar in its V3 loop to subtype C, though not identical [51]. These genetic features could explain the rarity of X4 variants in subtype C or subtype A infection. On the other hand, subtype D has been reported to be more X4-tropic, or exhibit dual (CXCR4/CCR5) usage in some cases [49,52,53,54]. The V3 loop of subtype D viruses is identical to R5-tropic viruses, suggesting other regions outside of the V3 loop affect CXCR4 usage for subtype D [52]. It has been shown that the majority of the latent reservoir in subtype B encompass viruses that use CCR5 as co-receptor [55]. However, it is unclear whether this is true for other subtypes. Furthermore, the tropism of the virus can influence the subset of CD4 T cells in which HIV can establish the latent reservoir. Different subsets of CD4 T cells respond differently to different LRAs [56,57]. Further research will be needed to characterize whether the differences in tropism influences the establishment of latency in different subsets and how this could influence cure strategies.

## 4. The Viral Long Terminal Repeat (LTR)

Upon integration, the HIV-1 LTR becomes the HIV-1 promoter and recruits host transcription factors for viral transcription [58]. There are numerous mechanisms that contribute to HIV-1 latency, one of which is lack of activated host transcription factors such as nuclear factor kappa light chain enhancer of activated B cells (NF-κB), nuclear factor of activated T cells (NFAT), or signal transducer and activator of transcription 5 (STAT5) in resting memory CD4 T cells [59,60,61,62,63]. One of the rationale for latency reversing strategies is to provide these activated transcription factors in order to promote viral transcription.

The presence and number of transcription factor binding sites vary depending on the subtype (Figure 2), which in some cases has been shown to alter their transcriptional activation in response to diverse stimuli in vitro [64,65,66,67,68,69,70,71,72]. These studies are described below and are also summarized in Table 2. Of note, the subtype C LTR harbors more NF-κB binding sites than other subtypes, with 3–4 NF-κB sites [70,73] compared to all other subtypes which typically harbor 2. Subtype AE is an exception to this, as it harbors only one NF-κB binding site [74]. Though these additional κB sites have not been shown to directly compete for NF-κB binding in subtype C [68], their presence has been associated with greater transcriptional activity in response to NF-κB activating stimuli [73]. This was shown in an LTR-reporter system in Jurkat cells, where viruses with 4 NF-κB binding sites produced higher amounts of viral transcripts, compared to viruses with only 3 [73]. This phenotype was further confirmed in a subtype-specific LTR-luciferase reporter system in Jurkat cells, where stimulation with NF-κB-activating stimuli induced more luciferase expression in subtype C LTRs which had 4 NF-κB sites versus those that had 3 [64]. If they are bonafide NF-κB binding sites, these additional NF-κB sites could potentially result in increased transcription by recruitment of p65/p50 heterodimers; however, it could potentially impute a more silent phenotype in latent cells, through binding of the p50 homodimer NF-κB and recruitment of inhibitory complexes such as histone deacetylases (HDAC) to the LTR [75].

Rodriguez et al. characterized the LTRs of five subtype A and seven subtype C circulating strains in India. Sequence analysis showed that subtype C isolates contained three NF-κB sites, and there were no significant differences in transcriptional activity between subtype A and subtype C [67]. Transcriptional activity was determined by transfecting LTR-luciferase reporter plasmids into Jurkat cells and measuring luciferase in the absence or presence of HIV-1 Tat. One limitation of this study is that subtype B Tat was used, instead of the subtype-specific Tat, which could explain why they observed no differences in transcriptional activity. In a study by Roof et al., subtypes B, C, or E LTRs and subtypes B, C, or E Tat were transiently transfected into Jurkat cells and then stimulated with tumor necrosis factor-α (TNF-α). Luciferase activity was measured 10 hours post-stimulation. In this system, differences in basal level transcription were observed among subtypes, and subtype E Tat was found to have the highest transactivation capacity regardless of LTR. [68]. Most major subtypes (A–G) harbor a signal transducers and activators of transcription (STAT) binding site, though their exact sequence and intensity of response to activated STAT5 stimulation vary [65]. The location and number of activator protein-1 (AP-1) and specificity protein-1 (Sp1) sites vary as well, depending on the subtype [66,69,71]. The variance in Sp1 and NF-κB sites in the LTR of the Chinese subtype B’ was associated with a greater capacity to drive gene expression in a reporter system [72]. Jeeninga et al. examined the contribution to transcription by each of the subtype-specific LTR sequences (A, B, C1, C2, D, E, F, G and G’’). This study found that although there was no difference in tat-transactivation among subtypes, the response to TNF-α was correlated with the number of NF-κB sites present in the various viral LTRs [66]. Subtype C, which harbored three NF-κB sites, exhibited the greatest response to TNF-α. Using subtype-specific LTR-luciferase constructs, they found that subtype E had higher basal transcriptional activity. The core of the subtype E promoter was subsequently cloned into a subtype B molecular clone (HIV-1 LAI) and was used to infect SupT1 cells. This subtype B/subtype E chimeric virus produced higher CA-p24 than subtype B LAI, suggesting that genetic differences in subtypes may partly drive differences in viral replication kinetics [66].

The impact of host cell environment and subtype on viral replication was evaluated by van Opijnen et al. By employing viruses with subtype-specific LTRs (subtypes A, B, C, D, E, F, G) in a subtype B LAI background, they examined replication rates in SupT1 and MT2 cell lines in the presence or absence of phytohemagglutinin (PHA) or TNF-α (mimicking a resting state or activated state in T cells). They tracked viral replication by measuring CA-p24 across seven days of infection. The authors also assessed viral fitness by performing competition experiments where two distinct subtypes of virus were used to infect SupT1 cells. Infected cells were sequenced in order to determine the frequencies of each of the subtypes. They found that subtype E was the most fit but that this fitness changed depending on the cellular environment [69]. These studies highlight the importance of the host’s cellular environment in concert with viral-intrinsic factors such as subtype-specific LTR and suggest that some subtypes may be more adaptable to different cellular environments than others. Whether this difference also extends to latency dynamics has not been fully characterized.

Thus far, there have been limited studies evaluating the ability of non-B subtypes to establish latency in T cell lines [76,77,78]. These studies have been described in detail below, as well as summarized in Table 2. Van der Sluis et al. characterized the latency profiles established by chimeric viruses containing subtype A, B, C1, C2, D, AE, F, G, or AG specific LTRs with a subtype B LAI backbone. SupT1 cells were infected with these clone variants and cultured in the presence of a fusion inhibitor to prevent further infection. Latency was measured 24 hours later after stimulation with TNF-α by assessing CA-p24 by flow cytometry and p24 release by an enzyme linked immunosorbent assay (ELISA) [76]. Using this acute HIV-1 model system, the authors found no differences in latency among subtypes with the exception of subtype AE, which had higher basal transcription and low responsiveness to TNF-α. This phenotype was proposed to be related to the presence of a GA-binding protein (GABP) site in the place of an NF-κB site within the subtype AE LTR [76].

Duverger et al. further explored how a subtype-specific LTR sequence may affect the establishment of latency. HIV-1 clones which had LTRs from subtypes A, B, C1, D, E, F, and G were cloned into HIV-1 LAI and used to infect the GFP reporter T cell line J2574. This cell line contains a subtype B NL4-3 LTR that controls the expression of GFP either in response to HIV-1 activating stimuli or HIV-1 tat. GFP was used as a surrogate of tat expression and transcriptional activity. Interestingly, they found that subtype A and subtype C LTR viruses generated higher levels of latent infections (assessed by phorbol myristate acetate or PMA stimulation of day 23 cultured cells), and the subtype E LTR virus had a reduced ability to establish latency [77]. Using transcription element search software, the authors found an AP-1 binding site that was extended by seven nucleotides in subtypes A and C, and absent in subtype E LTRs. They confirmed the relevance of this site by deleting the AP-1 site from NL4-3 and assessing latent infection in the J2574 cell line. Unexpectedly, the authors also found that viruses with a subtype B LTR or subtype A LTR responded similarly to TNF-α, hexamethylene bisacetamide (HMBA, a cell differentiating agent), suberoylanilide hydroxamic acid (SAHA, a histone deacetylase inhibitor), and trichostatin A (TSA, a histone deacetylase inhibitor). So, although subtype A generated a higher level of latent infections, the percentage of cells that could be reactivated was similar to subtype B even with strong stimuli such as TNF-α, suggesting that perhaps there are other mechanisms involved in maintaining latency in subtype A apart from epigenetic modification or NF-κB activation.

Recently, Dahabieh et al. used a doubly fluorescent virus system to distinguish between immediate silencing leading to latency (LTR-silent), or progressive latency due to epigenetic silencing [78]. The Red-Green-HIV-1 (RGH) virus encodes both enhanced green fluorescent protein (eGFP) and mCherry, a red fluorescent protein. The expression of mCherry is under a cytomegalovirus promoter and is constantly on, while eGFP is controlled by the HIV-1 LTR and can be packaged into virions during active infection. Thus, cells that are productively infected are either yellow (due to the combination of red and green) or green, cells that are latently infected are red, and uninfected cells will not express either fluorescent protein. Subtype-specific LTRs (subtypes B, A, C, D, AE, F, and G) were cloned into the RGH vector and used to infect Jurkat cells. The authors found that the ability to establish latency was different among subtypes, where subtype D and subtype F were more prone to LTR silencing than other subtypes, whereas subtype AE was less prone to latency.

Taken together, these studies highlight differences in establishment of latency that are attributable to the viral LTR. Since these studies only replace the LTR and keep all other subtype B viral proteins, this effect may be more pronounced in a system with complete subtype-specific genomes. The general consensus from these studies is that subtype AE is more transcriptionally active and less latent. This may be due in part to the GABP site present in the subtype AE LTR. Most of the studies to date have been done in cell lines. As such, further studies are warranted to evaluate whether these differences are also observed either in primary cell models or in cells isolated from HIV-infected participants ex vivo.

## 5. HIV-1 Viral Proteins and Latency

The role of HIV-1 viral proteins in HIV-1 pathogenesis and immune evasion has been extensively studied with mostly subtype B, and these studies have been extended to non-B subtypes and reviewed elsewhere. Here we review and discuss the potential role of HIV-1 proteins in the establishment or reactivation of latency.

Once HIV-1 is integrated into the host genome, the specific site of integration can have a dramatic impact on the persistence of latently infected cells. This can occur, for example, by integration into genes that are typically associated with cancer or cell cycle such as *BACH2* or *MKL2* [79,80]. These previous studies associating the integration site and persistence of the latent reservoir were likely with subtype B viruses due to the geographic location of the studies, and to our knowledge, this has not been extended to other subtypes. The genetic diversity of the HIV-1 integrase protein has obvious implications when it comes to integrase-inhibitor resistance development [81,82], but whether this leads to differential preference in the integration site has not been shown. The sequence of integrase is mostly (60%) conserved across subtypes in key catalytic domains to preserve general functionality, though there are several regions of the protein that are less conserved (extensively reviewed in [83]). In a study by Demeulemeester et al., naturally occurring HIV-1 integrase variants were found among subtypes A, B, C, D, F, and G. The study found that the variant IN_S119G_ re-targeted viral integration from gene-rich areas to gene-poor regions, characterized by heterochromatin and H3K9 bi and trimethylation. The abundance of these variants was disproportionate among subtypes, where the global re-targeting variants IN_S119G_ occurred most in subtype B, and IN_R231G_ in subtype C. Interestingly, these variants were associated with faster disease progression in a chronic infection subtype C cohort [84]. Since the prevalence of these integrase variants differs among subtypes in this study, this could conceivably affect integration site patterns from subtype to subtype, thus impacting latency through this mechanism. However, these findings have not been confirmed by others and represent rare variants of the integrase protein. Further work needs to be done to determine whether the integration site differs due to subtype, or if the findings with subtype B integration sites in cancer-associated genes can be extended to other subtypes.

During infection, subversion of the host’s immune system is vital to the virus’ survival. As such, HIV-1 has numerous viral proteins to achieve this immune evasion. HIV-1 viral infectivity factor (Vif) is a viral protein known to counteract host antiretroviral protein apolipoprotein B mRNA editing enzyme catalytic subunit 3G (APOBEC3G) [85]. APOBEC3G deaminates viral cDNA which causes premature stop codons or amino acid substitutions, rending the virus defective. Vif counteracts this host restriction factor by targeting it for degradation, thus preventing APOBEC3G antiviral functions [86]. Lisovsky et al. showed that subtype-specific Vif function affected viral replication in the presence of APOBEC3G in a Tzm-Bl luciferase reporter assay [87]. Subtypes B, C, and AG Vif sequences were used to complement subtype-specific ΔVif virus backbones and transfected into 293T cells in the presence or absence of APOBEC3G. The resulting virus generated was used to infect Tzm-Bl cells, and viral replication was measured using luciferase. Interestingly, subtype C was less functional and more sensitive to APOBEC3G-mediated viral inhibition, compared to subtype B or subtype AG in this system. Since subtype C Vif was not as effective at counteracting APOBEC3G, the viral genome was hypermutated by APOBEC3G, thus limiting the ability of the virus to infect and subsequently replicate. Because most of the latent reservoir found in patients contain defective and hypermutated proviruses [88], it is tempting to speculate that subtype-specific vif could influence the establishment of an intact and inducible latent reservoir.

The HIV-1 viral protein R (Vpr) is involved in the transport of the pre-integration complex to the nucleus, modulation of cell death, cell cycle arrest, and LTR-transctivation [85,89]. Vpr has been found highly conserved in subtype B latent proviruses isolated from ART-treated individuals, suggesting a selective advantage to preserve the *vpr* reading frame [90]. However, whether this plays a role in HIV-1 latency is not clear. Regarding its transactivation potential, subtype B has been shown to more potently transactivate the HIV-1 LTR than subtype C [91]. However, how this influences either the establishment of latency or reactivation has not been characterized.

HIV-1 viral protein U (Vpu) also plays a critical role in downmodulating host immune responses against HIV-1, such as counteracting Tethrin and downregulating CD4 [85,92]. Recently, it has been shown that Vpu potently inhibited NF-κB at the transcriptional level and prevented immune responses against HIV-1 [93]. In this study, subtype C and subtype B viruses, which were either wildtype for Vpu or lacked Vpu, were used to infect primary CD4 T cells. Infected cells were subjected to RNA-Seq. Transcription factor network analyses showed suppression of NF-κB target genes in the presence of Vpu, suggesting a role of Vpu in the establishment of latency.

Nef is one of the most studied HIV-1 accessory proteins, likely due to its prominent role in the host immune response to HIV-1 and pathogenesis [85]. An early study examined 41 viruses which represented subtypes A, B, C, E, F, G, and H and characterized their Nef sequence. Through phylogenetic analysis, they found that the distance between strains within the same subtype to be 9.6%, and the distance between subtypes to be 20.4% [94]. The impact of these sequence variations so far has been evaluated in vitro. Mann et al. examined over 300 single HIV plasma RNA Nef clones from ART-naïve chronically infected patients that were almost equal parts subtypes A, B, C, or D. They transfected cloned Nef sequences into an immortalized CD4 T cell line and measured CD4 and human leukocyte antigen (HLA) class I surface expression by flow cytometry. Interestingly, they found that with regard to CD4 and HLA I downregulation, subtype B was the most functional, followed by subtypes A or D, and lastly subtype C [95]. The authors also found substantial inter-subtype sequence diversity, which could potentially explain inter-subtype functional differences. In another publication, Mann et al. explored the underlying mechanisms of this functional difference between subtypes and examined genetic determinants of Nef function in subtypes B and C, the most functionally disparate subtypes [96]. Using the same experimental setup of transfecting Nef sequences into an immortalized T cell line and examining CD4 and HLA class I surface expression by flow cytometry, they mutated specific residues in NL4-3 (subtype B) Nef to examine the impact of subtype-specific mutations on Nef function. They found sequence variations at codons 20 and 88 significantly affected Nef’s ability to downregulate CD4 and HLA-I [96]. Mutations at codon 88, specifically S88G mutation, is significantly more frequent in subtype C viruses, which could partially explain the functional differences observed between subtypes. Nef has also been shown to reactivate latent HIV from latency in cell lines [97,98,99], but whether these subtype-specific differences directly impact latency is not well established.

There are other experimental evidences suggesting that some accessory genes may play a role in HIV-1 latency, though these studies have been mostly with subtype B. Recently, the group of Susanna Valente has reported that mutations in both Nef and Vpr proteins that enhance their ability to activate NF-κB conferred resistance to the Tat inhibitor didehydro-cortistatin A (dCA) [100]. dCA is an alkaloid shown to repress HIV-1 reactivation from latency and promote long-term latency [101,102]. Furthermore, the presence of accessory genes may influence the establishment of latent infections in a primary cell model that uses fetal thymocytes [103]. The genetic sequence of accessory proteins can vary from 5% to 35% by subtype, potentially affecting function [104,105]. Thus, we speculate that perhaps non-B viral proteins may have varying degrees of impact on HIV latency. Further studies using non-subtype B viruses are needed to address this possibility.

## 6. The Size of the Latent Reservoir

Most of the studies evaluating the size, long-term dynamics, and composition of this latent reservoir in vivo have been done in subtype B individuals, though a few recent studies have examined the reservoir size in non-B participants. For a summary of these studies thus far, see Table 3. Two of these studies have examined the size of the replication-competent, inducible latent reservoir in ART-treated virally suppressed non-B individuals. In the first study, Prodger et al. examined the size of the latent viral reservoir in a Ugandan cohort and compared it to a previously published Baltimore cohort. Interestingly, they found that long-term virally suppressed individuals with non-B subtypes (subtypes A, D, and A/D recombinant) had a 3-fold lower frequency of inducible latently infected resting CD4 T cells as measured by the quantitative viral outgrowth assay (QVOA) [106]. The authors could not attribute this difference to subtype due to the Ugandan cohort having subtypes A, D, or A/D while the Baltimore cohort was exclusively subtype B. The majority (69%) of individuals in the Ugandan cohort were subtype D, which was shown to be more prone to latency than other subtypes in an in vitro model of latency [78]. This could be a potential explanation as to why the Ugandan cohort had a lower measured reservoir since the QVOA involves reactivating the cells in vitro. Alternatively, an unknown mechanism that may involve viral-intrinsic or host-intrinsic factors may be influencing the establishment of latency in this population.

In the second study, Omondi et al. assessed the viral reservoir in a smaller cohort of early-treated individuals (subtypes B, AE and G) and found a correlation between subtype-specific Nef function and reservoir size [107]. This study measured the size of the latent HIV-1 reservoir using QVOA from acute/early treated individuals and also cloned viral Nef sequences to assess Nef function using an in vitro flow cytometric assay. They found that subtype B Nef was most effective at downregulating CD4 and HLA. These results are in line with the previous observation that subtype B Nef is most functional compared to other subtypes [95,96]. The authors reasoned that Nef function could be allowing infected cells to evade HLA-mediated immune responses, thus explaining the larger reservoir size of subtype B individuals in this cohort. However, the complete mechanism underlying this phenotype of a reduced size of the latent reservoir could not be solely attributed to Nef function [107]. The results from these studies can be interpreted in one of two ways. First, individuals with non-B subtype viruses potentially establish latency at a reduced frequency than subtype B, due to differential immune responses or functionality of viral proteins. Perhaps making the argument that subtype B is more evolutionarily fit to persist through viral latency in unsuitable cellular conditions. Second, it could be that non-B subtypes achieve a deeper state of latency, perhaps due to changes in integration sites, methylation of viral promoter, or LTR sequence, making them less inducible than subtype B. Further work needs to be done to understand the mechanisms behind this reduction of inducible reservoir observed in these two studies.

In the third study, the long-term dynamics of the viral reservoir were measured in non-B individuals. In a large cohort study of 1057 ART-suppressed individuals, Bachmann et al. evaluated host and viral characteristics and whether any of these variables were associated with reservoir size [108]. The viral reservoir was determined by total HIV-1 DNA after 1.5 years of ART initiation. The initiation of ART within a year of infection was associated with the size of the reservoir. This finding confirms other studies where this was also observed [109,110]. They found that decay of the latent reservoir was significantly faster in non-B individuals, compared to subtype B individuals in a multivariable model. HIV-1 subtype was only associated with a smaller reservoir in univariable analysis, and this significant difference did not persist in multivariable analysis, though there was a trend. Interestingly, this study also found that ethnicity affected reservoir size. Non-white individuals were associated with a lower reservoir size. As distinct subtypes circulate in separate regions of the world, ethnicity is an important variable to consider. Using a Wilcoxon rank sum test, the authors found that subtype C individuals had lower total levels of HIV-1 DNA than subtype B, and that this difference was still observed after controlling for ethnicity.

To our knowledge, these are the only studies comparing the size of the reservoir of non-subtype B to subtype B. Though the studies are quite dissimilar in cohort sizes, subtypes included, and cohort type (long term vs. early treated), two of the three report the same finding: non-B individuals have a reduced frequency of latently infected cells that harbor replication-competent virus. The third study found a trend towards this phenotype in their cohort of individuals who were on ART for 1.5 years, suggesting that subtype was not associated with reservoir size. One potential explanation for this opposing conclusion could be that the polymerase chain reaction (PCR)-based assays and the QVOA used to measure the reservoir do not correlate very well [111]. Total HIV-1 DNA may over-estimate the size of the latent reservoir compared to the QVOA [111]. The large majority of integrated proviruses are defective [112,113], which are detected in PCR assays but not in the QVOA. The QVOA assay has drawbacks as well, as it can underestimate the true size of the reservoir, as a subset of these non-induced viruses are intact and indeed replication-competent [112]. Clearly, the development of better methods to measure the latent reservoir, in addition to expanding our understanding of the dynamics affecting the latent reservoir in non-B individuals, is greatly needed. With that said, if non-B subtype is indeed associated with smaller reservoir size, this can have dramatic implications for cure strategies, as a cure strategy may be more feasible with individuals who have smaller latent reservoirs. Additionally, understanding the mechanisms that are responsible for this reduction can help with efforts to reduce or preclude the establishment of latency.

The complete mechanisms that affect the size of the latent reservoir in both subtype B and non-B infections are incompletely understood but there are several known factors that can contribute, such as the lack of host transcription factors, the lack of viral tat, and epigenetic silencing (reviewed in [5]). Furthermore, it is well accepted that the timing of initiation of ART can also limit the establishment of the latent reservoir [109,110,114,115]. The host immune response has also been associated with the size of the latent reservoir. It has been shown that there is an association between early immune responses and the size of the latent reservoir in a small cohort of early treated individuals [114]. Yue et al. found that granzyme B T cell responses against Tat/Rev, Env, Gag, and Vif negatively correlated with HIV-1 proviral loads and size of the latent reservoir (measured by QVOA) at both the beginning of the study and 48 weeks [114]. These findings suggest that initial immune responses to HIV-1 may prevent the establishment of the latent reservoir. Though the subtype of the individuals was not included, study participants were recruited in Toronto, Ontario, Canada and are likely to be subtype B. To our knowledge, these observations on immune responses and reservoir size have not been extended to other subtypes; however, there is some limited data on the quality of CD8 T cell responses in non-B subtypes. In a study by Baker et al., CD8 T cells from 56 HIV-1-infected participants (which were either subtypes A or D) were stimulated with consensus subtype A or D Gag peptides ex vivo. The subsequent CD8 T cell response, defined in this study by Interferon-γ (IFN-γ) production and CD107a/b (a degranulation marker), was measured by flow cytometry. The authors found that IFN-γ production was not significantly different between the two subtypes but observed a significant reduction in CD107 expression in individuals infected with subtype D compared to subtype A. These data suggest that distinct subtypes can elicit varying CD8 T cell responses. The molecular mechanisms underlying this difference in immune response have been somewhat evaluated in vitro. 

There is also some limited in vitro evidence that the nucleotide composition of HIV-1 affects type-I Interferon responses, as HIV-1 has a different average composition of amino acids than human sequences. Vabret et al. found that nucleotide divergence was correlated with type-I Interferon response. Among subtypes A, B, C, D, F, and G, subtype D had the highest nucleotide divergence, which was correlated with subtype D’s increased pathogenicity [116]. Thus, the general composition of nucleotides in HIV-1’s genome may play a part in immune responses. In addition to nucleotide composition, viral protein function can also vary by subtype [47,87,91,95,117,118,119]. Because of these two factors, one can speculate that the immune responses generated against different subtypes may vary, potentially impacting the generation of the latent reservoir through this mechanism. The relationship between non-B subtypes, immune responses elicited against them, whether or not they differ in quality or quantity, and the size of the latent reservoir has yet to be proven. This area of research requires further investigation to understand HIV-1 latency in the non-B subtype, which is extremely relevant as the majority of the worldwide HIV-1 infections are with non-B subtypes [18,21,22,23].

## 7. Latency Reversal—The “Shock”

“Shock and kill” strategies are based on eliciting viral gene expression from latently infected cells using an LRA, thus allowing for the elimination of latently infected cells by virus-mediated cytopathicity or immune-mediated removal by NK or CD8 T cells. LRAs are small molecules that can induce viral gene expression in latently infected cells (for reviews in LRAs see [120,121,122]). Thus far, the pre-clinical screening and development of LRAs have used in vitro systems with subtype B HIV-1 viruses. There is an assumption that pre-clinical LRAs that are effective with subtype B will be just as effective for all other subtypes. However, given that there are substantial sequence differences in LTR [66,67,68,69,76,123,124], and over 80% of the world is not subtype B [23], this is an area of research that requires further attention. HIV-1 cure strategies may require subtype-specific approaches if they are not universally applicable. Below, we summarize several different classes of LRAs that have been developed thus far, and in what contexts they have been tested relevant to different HIV-1 subtypes. These studies are also summarized in Table 4.

### 7.1. Protein Kinase C (PKC) Agonists

PKC agonists mainly activate the transcription factor NF-κB. There are several isoforms of PKC, and their regulation can be independent or dependent on calcium and diacylglycerol [125]. PKC agonists can activate distinct isoforms of PKC, ultimately activating NF-κB but through slightly different ways. Under basal conditions, NF-κB is complexed with IκBα in the cytoplasm and this prevents NF-κB’s transcriptional activity (reviewed in [126,127]). Activation of PKC induces signaling that allows release from this inhibitory complex, allowing NF-κB to be activated and translocate to the nucleus. In the setting of HIV-1 infection, NF-κB can be recruited to NF-κB binding sites present in the viral LTR, thus appropriating NF-κB to promote its own transcription. There are several PKC agonists developed thus far, some of which have advanced to clinical trials (extensively reviewed in [15,121]).

The success of using NF-κB activation as a strategy to reactivate latent HIV-1 is impacted by whether the NF-κB binding sites exist in the LTR. Subtypes A, B, and D harbor two, subtype AE harbors one, and subtype C typically harbors three to four NF-κB binding sites [66,73]. The effect that these additional NF-κB sites has on latency reversal has been somewhat evaluated in cell lines, with a first study finding no difference in latency reversal with the exception of recombinant subtype AE [76] and a second study finding differences with several subtypes [71]. 

Prostratin is a non-tumor promoting phorbol ester that reactivates in both cell lines and ex vivo patient cells [128,129]. Prostratin, alone or in combination with the HDAC inhibitor vorinostat (also known as SAHA) was examined in one study across several subtypes [71]. Using both a lentiviral latency system and a primary cell latency model, Burnett et al. explored the latency profiles of subtypes A, A2, B, C, C’, D, F, H and recombinant subtypes A/G, B/C and B/F [71]. Using the LTR-GFP-IRES-Tat (LGIT) virus system in Jurkat cells, the study evaluated single and combinatorial latency reversal strategies involving PKC agonists and epigenetic drugs. Interestingly, subtype C exhibited the weakest response, while subtypes A and A/G, B/F, and F exhibited strong responses. These studies were extended to a more biologically relevant cell type, primary CD4 T cells, using the same LGIT virus system, and similar patterns were observed. These data suggest differences in the magnitude of responses that are solely attributable to subtype-specific LTR differences as the LGIT virus does not differ in other viral genome elements, and that some subtypes may be more amenable to these single or combinatorial agents for latency reversal. 

In the last decade, other PKC agonists have been described, such as bryostatin-1 [130], ingenol dibenzoate [131,132], and ingenol mebutate (also known as Ingenol 3-angelate) [133,134]. Other semi-synthetic ingenols have been developed [64]. These agonists have been shown to be potent reactivators of latent HIV-1 with subtype B but have not been evaluated against non-subtype B viruses, with the exception of a study with ingenol B. Jurkat cells were transfected with subtype B and subtype C LTRs (some of which had 3 NF-κB or 4 NF-κB sites) and stimulated with various PKC-activating compounds such as TNF-α, PMA, or ingenol B. They observed that subtype C LTRs with 4 NF-κB sites had the greatest response. However, this study was limited to cell lines and transient transfection of LTR constructs. Further work needs to be done in other systems that better recapitulate latency in order to confirm these findings. With that said, this study highlights how the presence of an additional NF-κB site can enhance the response to NF-κB-activating stimuli.

All these PKC agonists activate the general PKC-NF-κB signaling axis. So, it can be speculated that some subtypes may be more amenable to latency reversal through the PKC-NF-κB signaling axis. Further studies are necessary to confirm these results using primary cell models and ideally latently infected cells from well-suppressed HIV-1-infected participants.

### 7.2. Toll-Like Receptor Agonists

Toll-like receptors (TLRs) are pattern recognition receptors that play critical roles in detecting microbial pathogens [135]. They are localized at either the cell membrane (TLR-1, -2, -6, -4, -5) or in endosomes (TLR-3, -7, -8, -9) (for a review, see [16,136]). Our lab has shown that TLR-1/2 agonist Pam3CSK4, TLR-2/6 agonist Pam2CSK4, and dual TLR-2/7 agonists reactivate latency in a primary cell model of latency and ex vivo patient cells in a NF-κB-dependent manner [137,138]. MGN-1703 is another TLR agonist that reactivates latent HIV-1 through TLR-9 [139]. There have been several exciting LRAs under this class that have moved onto clinical trials, one of which being GS-9620 (vesatolomid). GS-9620 is a TLR-7 agonist that has been shown to reactivate latent HIV-1 from aviremic participants and improve effector cell functions [140]. Currently, clinical trials are underway to determine the safety of GS-9620, in addition to other secondary measures such as time-to-rebound following an analytical treatment interruption (NCT03060447, NCT02858401). Due to the geographical location, the clinical trial participants will likely be subtype B. To the best of our knowledge, TLR ligands have not been evaluated against non-B subtypes although, making the same speculation based on NF-κB sites, there may be varying degrees of efficacy using this class of LRAs.

### 7.3. Non-Canonical NF-κB Agonists

NF-κB is part of a family of transcription factors that can form hetero or homo dimers composed of RelA/p65, c-Rel, RelB, p50 or p52 subunits. The composition of NF-κB has differential effects on transcription [126]. Homodimers of p50 or p52 can act as transcriptional repressors since they do not contain a transactivating domain, which is known to be critical for transcriptional activity. Second mitochondrial-derived activator of caspases (SMAC) mimetics mimic the tetrapeptide region of SMAC, which binds to the baculoviral inhibitor of apoptosis protein repeat (BIR) domain. These BIR domains are present in inhibitor of apoptosis (IAP) family member proteins, including the baculoviral IAP repeat containing-2 (BIRC2) protein. This binding to BIRC2 prevents the degradation of the NF-κB inducing kinase (NIK), allowing for the accumulation of NIK and induction of p100 processing into p52 [126]. Using an RNA interference screen in HEK293T cells using vesicular stomatitis virus-pseudotyped NL4-3 (a subtype B virus), it was found that BIRC2/cIAP1 negatively regulates HIV-1 transcription and its pharmacological inhibition reactivated latent HIV-1 [141]. They found that this reactivation occurred through the non-canonical NF-κB pathway. Sampey et al. found that AZD5582, a SMAC mimetic, also reactivated latent HIV-1 through the non-canonical NF-κB pathway. AZD5582 has been tested in ex vivo resting CD4 T cells from aviremic participants [142] and in vivo mouse models and non-human primates [143]. To the best of our knowledge, this has not been tested with other HIV-1 subtypes apart from subtype B, and SIVmac239. These compounds may also have varying effects on viral reactivation, likely dependent on the subtypes evaluated.

### 7.4. STAT Modulators

The study of STAT binding sites across subtypes has been somewhat limited. Crotti et al. examined the LTR of subtypes A, B, C, D, F, G, AE, AG, and found differences in putative STAT binding sites. Further, they found that in a U937 reporter system, the subtype G LTR was most responsive to phosphorylated STAT5, the activated form of the transcription factor [65]. This could be pertinent to the development of HIV-1 cure strategies which employ STAT activation as a method of reversing latency [63], as this study suggests some subtypes would be more responsive to STAT5-mediated reactivation. Thus far, STAT modulators have been tested in subtype B contexts, in vitro. Further studies are needed to confirm whether different subtypes possess the same sensitivity to latency reversal through the STAT5 signaling pathway.

### 7.5. Epigenetic Modifiers and PTEF-b Release Agents

Epigenetic modification is another critical factor of viral latency (reviewed extensively in [5,144]). There are several types of epigenetic modifiers, including bromodomain and extra terminal (BET) inhibitors, DNA and histone methyltransferase (DMT, HMT, respectively) inhibitors, and HDAC inhibitors. Many of these epigenetic modifiers have anti-cancer properties [145,146,147] in addition to their HIV-1 latency-reversing activity [148,149,150,151,152,153,154,155,156].

Repressive epigenetic marks such as methylation can be alleviated with HMTs, and lack of chromatin accessibility can be remedied with HDAC inhibitors. These strategies have been shown to be effective for reversing latency in both in vitro latency models and ex vivo HIV-1-positive participant cells [150,151,152,153]. Within HDAC inhibitors, some drugs can affect all HDACs (pan-HDAC inhibitors such as SAHA), as well as selective class 1 HDAC inhibitors like MS-275 (entinostat). Entinostat has been evaluated as a latency-reversing agent (either as a single agent or in combination with other LRAs) in latent cell models [153].

Thus far, the HDAC inhibitor vorinostat (SAHA) and valproic acid have advanced the farthest in clinical trials, but studies have shown it has not had an impact on the size of the latent reservoir in clinical trials as a single agent [152,157,158]. Importantly, these studies were conducted in Europe/United States, meaning that the study population was likely subtype B. DNA methyltransferas inhibitors such as 5’Azacytidine have been developed as LRA [150], though the role of DNA methylation in viral latency remains controversial [159,160]. Thus far, studies that have examined the efficacy of these epigenetic modifiers have been limited to subtype B, with the exception of Vorinostat [71]. Vorinostat did not meaningfully reactivate latent HIV-1 in both cell lines and primary cells alone, but in combination with Prostratin showed varying efficacy across subtypes A, A2, B, C, C’, D, F, H and recombinant subtypes A/G, B/C and B/F [71] (Table 4). Apart from this study, it has not been examined whether some subtypes are more or less sensitive to epigenetic modifiers either alone or in combination with other LRAs.

For HIV-1 latency reversal, repressive chromatin environments (methylated, lack of acetylation) can limit transcription initiation, but transcription initiation is not the only block for efficient viral reactivation [161]. Transcription elongation also has to occur for a successful reactivation. P-TEFb is normally sequestered in a transcriptionally inactive complex, the 7SK small ribonucleoprotein, and P-TEFb is critical for promoting elongation of transcripts through RNA polymerase II phosphorylation (reviewed in [162,163]). Release from this complex by small molecules or drugs allows P-TEFb to bind to HIV-1 tat to promote HIV-1 transcript elongation. However, there is competition for this binding as P-TEFb can also be bound by host BET proteins [164]. It has been shown that BET inhibitors such as JQ-1, can reactivate latency by releasing P-TEFb or interfering with BRD4 binding to P-TEFb, thus allowing for Tat binding [148,155,156]. To our knowledge, this class of LRA has not been tested in systems that use non-B subtype viruses. As Tat transactivation potential differs by subtype [68], it is possible that bromodomain inhibitors may be more effective in some subtypes than others. Further work is necessary to confirm whether this is true. 

### 7.6. Enhancing Viral Reactivation with Combination Treatments

LRAs that have been developed thus far are not exceptionally potent as single agents, compared to in vitro maximal activating stimuli such as T cell receptor engagement [165]. This issue can be potentially solved by combining distinct classes of LRAs to hopefully synergize and achieve the maximal viral reactivation without overt toxicity. Recent work by Laird et al. have identified a few combinations of LRAs that are effective at reversing latency in ex vivo HIV-1-infected participant cells, namely PKC agonists combined with bromodomain inhibitors or HDAC inhibitors [166]. There is limited data on whether the combination of LRAs could also be effective in non-B subtypes. A study by Burnett et al. included prostratin, a PKC agonist, and several epigenetic modifiers such as SAHA and HMBA. They found that the greatest responses were produced when the LRAs were used in combination, though not all subtypes were equally responsive to the combination [71]. Even so, the viral reactivation obtained from a combination was greater than single-agent treatment for the majority of subtypes.

Interestingly, checkpoint inhibitors have also been combined with LRAs to increased viral reactivation. It has been shown that checkpoint inhibitors alone have limited effects ex vivo on HIV-1 reactivation in HIV-1-infected participant cells [167]. However, the checkpoint molecule programmed cell death-1 (PD-1) plays a role in suppressing viral reactivation [168]. Fromentin et al. showed that PD-1 engagement inhibited viral production and blocked T cell receptor engagement-mediated viral reactivation in ex vivo HIV-1-infected participant cells. They further showed that combination treatment with an anti-PD-1 antibody and potent PKC agonist bryostatin-1, latency reversal is enhanced without deleterious effects of T cell activation [168]. In this study, the subtype of the participants is not denoted but are from countries in North America, so it is likely that the majority of participants were subtype B. These studies have not been extended to other subtypes, but it is a novel and interesting path to potentially enhancing latency reversal with the traditional LRAs. 

## 8. Future Perspectives

Despite the progress in our general knowledge of HIV-1 infection and HIV-1 latency, there still remains much to be explored with regard to the study of non-subtype B viruses in both model systems and in vivo studies. Thus far, it is clear that there are differences in viral pathogenesis, viral protein function, and some early studies in vivo showing that the size of the latent reservoir differs in subtype B from non-subtype B infected individuals. The mechanisms underlying the reduced reservoir in non-subtype B infected individuals in vivo are likely multifaceted. As the field makes advances towards developing more effective LRAs or combinations of LRAs, the matter of subtype should not be forgotten. Availability of infectious molecular clones of non-B subtypes can facilitate their study in in vitro systems for pre-clinical development of LRAs. Thus far, work in cell lines has given some credence to the idea that viral subtype may impact the establishment of latency or the ability to reactivate from latency. These studies should be extended to primary cells with viruses that encode all viral accessory genes and are replication-competent. For pre-clinical LRA development, primary cell models with non-B viruses can also be used to test efficacy, and ideally non-B ART-suppressed HIV-1-infected participant cells. Primary in vitro systems can also be used to investigate the mechanisms responsible for the establishment of latent infections. Further work needs to be done to show whether subtype is a factor in viral latency in order to develop a global and relevant cure strategy.

## Figures and Tables

**Figure 1 viruses-11-01104-f001:**
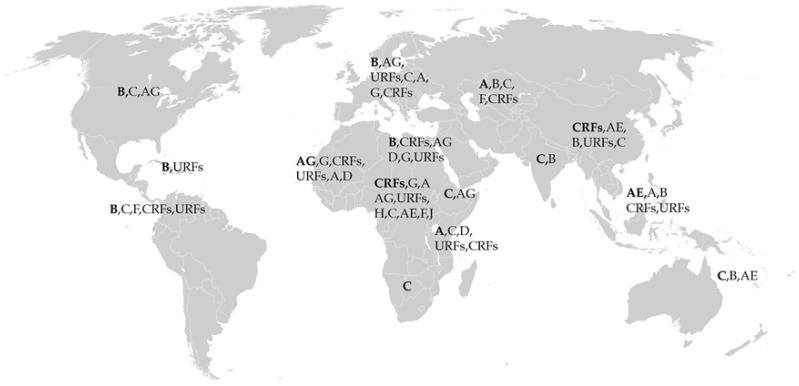
Global distribution of HIV-1 subtypes. The major subtype that circulates in distinct geographical locations are listed first and are in bold, followed by other reported subtypes in the region of interest. This depiction is based on estimates of circulating subtypes from 2004 to 2007 [18,21,22,23]. CRF: circulating recombinant form; this describes other CRFs which are not AE, AB, or AG. URFs: unique recombinant forms.

**Figure 2 viruses-11-01104-f002:**
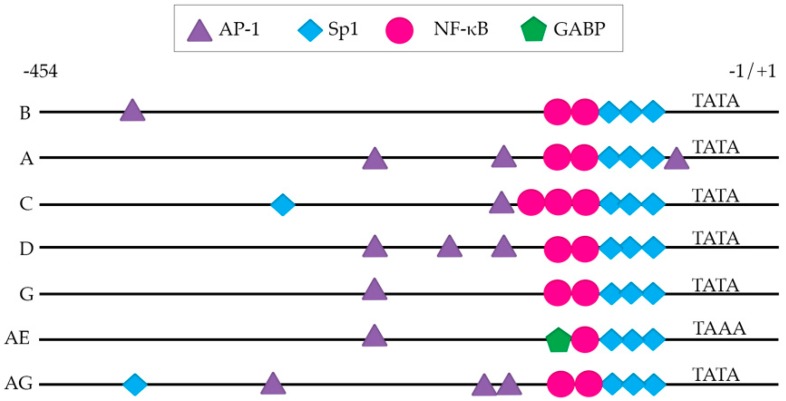
Long terminal repeat (LTR) from diverse subtypes. Main transcription factor binding sites in each LTR. Activated protein-1 (AP-1); specificity protein-1 (Sp1); nuclear factor kappa-light-chain-enhancer of activated B cells (NF-κB); GA-binding protein (GABP); TATA box promoter sequence for transcription initiation. Several other transcription factor binding sites have been identified and are not denoted in the graph for simplicity. This schematic is based on binding sites reported elsewhere [66,69,71].

**Table 1 viruses-11-01104-t001:** Summary of in vivo studies assessing the relationship between subtype and pathogenesis.

Geographical Location	Study Population	Study Size	Subtypes Included	Subtyping Method	End Points	Results	Reference
Israel	Ethiopian immigrants, non-Ethiopian Israeli men	168 (77 subtype C Ethiopian immigrants, 91 subtype B non-Ethiopian Israeli men)	C, B	V3 *env* peptide immunoassay, direct sequencing of V3 *env* region PCR	CD4 and CD8 counts	No difference in rates of CD4 decline between both groups	[30]
Senegal	Seronegative registered female sex workers	1683 seronegative enrolled, 81 seroconverted, 54 samples were subtyped	A, C, D, G	C2-V3 *env* region	AIDS-free survival, defined by <200 CD4 cells/mm^3^	Non-A subtypes were 8 times more likely to develop AIDS than A subtypes	[38]
Thailand	HIV-1 positive inpatients	2104 subtyped individuals	B’, E	V3 loop *env*-based peptide EIA	Total CD4 counts	No difference in immuno-suppression between subtypes	[29]
Sweden	HIV-1 infected outpatients	98 individuals (49 ethnic Swedes, 39 ethnic Africans)	A, B, C, D	V3 *env* sequencing	CD4 count, CD4 decline,	No association in disease progression or CD4 decline and subtype	[28]
Uganda	HIV-1 infected adults	1045 either A or D subtype individuals	A, D	Peptide serology, HMA	Progression to death, CD4 cell count	Subtype D associated with faster progression to death than subtype A	[33]
Tanzania	HIV-1 seropositive pregnant mothers	428 samples where subtype was determined	A, C, D, Recombinants	C2-C5 *env* region and 3’ p24/5’-p7 region of *gag*	Progression to death, WHO stage 4 clinical disease, CD4 cell count	Subtype D associated with the fastest progression to death, WHO stage 4 of illness, CD4 <200 cells/mm^3^ than subtype A or C	[35]
Kenya	HIV-1 seronegative commercial female sex workers	145 women	A, C, D	V1-V3 loops of *env,* HMA, sequencing and phylogenetic analysis	Mortality, CD4 counts	Subtype D associated with higher mortality and faster CD4 decline	[32]
Uganda	HIV-1 seroconverters	312 individuals	A, D, Recombinants, multiple	Multiregion hybridization assay	CD4 decline	Subtype D associated with faster CD4 decline than subtype A	[31]
Uganda	HIV-1 incident ART-naïve individuals	292 individuals	A, D, A/D, C, other recombinants	Partial *gag, env, pol* sequencing	CD4 ≤250 cells/mm^3^, WHO clinical stage 4 AIDS, death before and after ART introduction	Subtype D associated with faster disease progression than subtype A	[34]
Kenya, Rwanda, South Africa, Uganda, Zambia	Adult and youths with documented HIV-1 infection	579 individuals were subtyped	A, C, D	*pol* sequencing	CD4 count <350 cells/µL, viral load of 1x10^5^ copies/mL, clinical AIDS	Subtype C progressed faster than subtype A, subtype D progressed faster than subtype A	[37]
Sub-Saharan Africa (Uganda, Zimbabwe)	Newly infected HIV-1 women	303 women	A, C, D	PR, RT, and C2-V3 *env* region	CD4 decline	Subtype D was associated with faster CD4 decline, followed by subtype A, then subtype C	[36]

WHO: World Health Organization; PCR: polymerase chain reaction; HIV-1: human immunodeficiency virus-1; AIDS: acquired immunodeficiency; EIA: enzyme immunoassay; HMA: heteroduplex mobility assay; ART: antiretroviral therapy; PR: HIV-1 protease; RT: HIV-1 reverse transcriptase.

**Table 2 viruses-11-01104-t002:** Summary of in vitro studies that have assessed non-B subtypes and the establishment of latency.

Model System	Viruses Used	Subtype Assessed	Experimental Methods	Conclusions	Reference
C33A, HeLa, COS, U87, U373, SupT1	Subtype-specific LTR in a subtype B LAI background, subtype-specific LTR-luciferase reporter plasmid	A, C1, C2, D, E, F, G, G’’	Transfection of subtype-specific LTR- luciferase constructs to measure basal LTR activity and LTR activity in response to subtype B tat protein and TNF-α stimulation	Correlation between number of NF-κB sites and TNF-α response, subtype C had the greatest response Basal LTR activity in non-B subtypes significantly higher than subtype B, no difference in response to subtype B Tat transactivation Subtype-E LTR -LAI virus replicated better than subtype B-LAI	[66]
U937, Jurkat	LTR-luciferase reporter	B, C, E	Transfection of subtype-specific LTR-luciferase construct and addition of subtype B, C, E Tat to measure Tat transactivation, with or without TSA	Clade E Tat has the most transactivation activity Duplicated NF-κB sites in subtype C do not compete for NF-κB binding Basal activity of LTRs varied by subtype	[68]
SupT1, MT2	Subtype-specific LTR in a subtype B LAI background	A, B, C, D, E, F, G	Used CA-p24 ELISA to measure viral fitness/replication	Subtype-specific LTR impacts viral replication, viral fitness influenced by cellular environment	[69]
Jurkat, Jurkat Tat-T	Subtype-specific LTR luciferase reporter plasmid	A, C	Used transfection of LTR-luciferase in presence or absence of subtype B tat, to measure transcriptional activity	No difference in transcriptional activity between subtypes A and C	[67]
293T/U937	Subtype-specific LTR-luciferase construct	A, B, C, D, E, F, G, G’’	LTR transactivation studied by Co-transfection of subtype-specific LTR-luciferase construct and active pSTAT5	Potency of LTR transactivation by active STAT5 differs between subtype	[65]
Jurkat, Primary CD4 T cells	LTR-GFP-IRES-Tat (LGIT) virus with subtype-specific LTRs	A, B, A2, A/G, B/C, C’, C, B/F, D, F, H	Flow cytometry to determine LGIT-virus infected cells and latency reactivation with LRAs	Varying degrees of sensitivity to reactivation by single agent or combination of LRAs across subtypes	[71]
Jurkat, SUPT1	Subtype-specific LTR in a subtype B LAI background	A, B, C1, C2, D, AE, F, G, AG	Flow cytometry and ELISA for CAp24 for assessing reactivation	No differences except subtype AE and G were less prone to become latent, subtype AE had a significantly different response to LRA Vorinostat	[76]
Jurkat	Double-labeled (Red mCherry protein, Green eGFP protein) with subtype-specific LTR in a subtype B LAI background	A, B, C, D, F, G, AE	Flow cytometry to measure mCherry(red) expression and eGFP(green) expression. mCherry^+^eGFP^-^ cells are latently infected	Difference in degree of silent infections across subtypes, as well as sensitivity to PMA/ionomycin-mediated reactivation.	[77]
J2574	Subtype-specific LTR in a subtype B LAI background	A, B, C1, D, E, F, G	Flow cytometry: difference in %GFP between untreated infected population and PMA-treated infected population is latent population	AP-1 binding site in LTR is important for establishment of latency. Subtype E promoter lacks this site and has reduced ability to establish latency, Subtype A and subtype C exhibited greater latency establishment. LRAs reactivated similarly in subtype A compared to subtype B.	[78]
HEK293T, Primary CD4 T cells	Subtype-specific LTR in a HIV-eGFP/VSV-G single cycle infectious virus, LTR driven luciferase reporter and LTR-driven dual luciferase/renilla reporter	AE, B, B’, C, BC	measurement of luciferase/renilla as HIV-1 gene expression, measurement of GFP positive cells as a measure of viral reactivation	Blockade of Sp1 or NF-κB sites using dead Cas9 suppresses viral reactivation. HIV-1 B’ LTR had the greatest transcriptional activity when stimulated with TNF-α	[72]

Subtype E has been now re-classified as subtype AE. LTR: long terminal repeat. ELISA: enzyme-linked immunosorbent assay. AP-1: activated protein-1. GFP: Green fluorescent protein. pSTAT5: phosphorylated signal transducers and activators of transcription-5. TSA: trichostatin A. IRES: internal ribosomal entry site. VSV-G: vesicular stomatitis virus-G protein. Sp-1: specificity protein 1. NF-κB: nuclear factor kappa-light chain enhancer of activated B cells.

**Table 3 viruses-11-01104-t003:** Summary of in vivo studies of the latent viral reservoir that include non-B cohorts.

Cohort Size	Cohort Type	Subtype	Measure of Viral Reservoir	Conclusions	Reference
Ugandan cohort: 70 Baltimore cohort: 51	ART-treated, virally suppressed adults	A, D, AD, B	QVOA	Non-B individuals had a reduced frequency of latently infected cells, the mechanism underlying this observation could not be determined	[106]
30	ART-naïve men with acute/early HIV-1 infection	B, G, AE	QVOA	Subtype-specific Nef-mediated HLA downregulation correlates with reservoir size; HIV subtype is a statistically significant multivariable correlate of reservoir size.	[107]
1057	ART-treated long term suppressed individuals	B, AE, AG, A, C, D, F, G, Numerous recombinant forms	Total HIV-1 DNA	HIV-1 non-B subtype was associated with faster decay of reservoir in multivariable analysis; HIV-1 non-B subtype was only significantly associated with a smaller reservoir in a univariable model, trend observed in multivariable analysis	[108]

ART: antiretroviral therapy. QVOA: quantitative viral outgrowth assay. HIV-1: human immunodeficiency virus-1.

**Table 4 viruses-11-01104-t004:** Summary of in vitro testing of LRAs with non-B subtypes.

Model System	LRAs Assessed	Subtypes Tested	Reference
LTR-GFP-IRES-Tat (LGIT) virus-infected Jurkat cells and primary CD4 T cells	SAHA Prostratin HMBA, TSA, Valproic acid	A, A2, B, C, C, D, F, AG, BC, BF, H	[71]
Subtype-specific LTR in a subtype B LAI virus background infected SUPT1 or Jurkat	Vorinostat	A, B, C1, C2, D, AE, F, G, AG	[76]
LTR-luciferase reporter construct and tat transfection into U937, Jurkat cells	TSA	B, C, E	[68]
Subtype specific LTR in a subtype B LAI virus-infected J2574 cells	TSA SAHA HMBA	A	[78]

LTR: long terminal repeat. GFP: green fluorescent protein. IRES: internal ribosomal entry site. HMBA: hexamethylene bisacetamide. TSA: trichostatin A. SAHA: suberoylanilide hydroxamic acid.

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
