# Peer review of "HIV-1 Latency and Latency Reversal: Does Subtype Matter?"

_viruses, 2019, doi:10.3390/v11121104_

Round 1
Reviewer 1 Report
HIV-1 Latency and Latency Reversal: Does subtype matter? (review)
(Manuscript ID: viruses-628948)
Indra Sarabia, Alberto Bosque
In this review, the authors clearly explained the highly genetical diversity within HIV-1 subtypes. The pathogenesis, the coreceptor tropism, the LTR promoter sequence variations in the different HIV-1 subtypes are clearly defined. The role of viral proteins in latency, the reservoir size characteristics and latency reversal of these reservoirs among the HIV-1 subtypes are nicely described. Considering that most studies have been performed on subtype B viruses, this review illustrates the need for a better understanding of HIV-1 latency in all viral subtypes, in order to apply a worldwide therapeutic strategy.
A few minor points need to be addressed by the authors:
1) In line 102, the authors stated : “Subtype C and subtype A rarely undergo this switch; subtype C viruses favor CCR5 more extremely throughout infection than subtype A”. From lines 103 to 106, the authors explained the V3 loop variation in subtype C, however they did not explain it for subtype A.
2) The authors should mention the major LTR differences found within different HIV-1 subtypes such as the ones mentioned in their cited references 44 (STAT-binding sites), 45 (numbers of NF-kappaB sites), 47 (competition between NF-kappaB binding and Tat transactivation capacity), 49 (promoter architecture) and 50 (response to reactivation from latency).
3) The authors could add some information about HIV-1 integration site specificity and the efficiency of the viral immune response depending on the viral subtype.
4) Line 342: remove capital letter from “for a review”.
5) A figure or a table illustrating the review should be added (such as a scheme representing the LTR architecture of the different subtypes with the differences in transcription factor binding sites).
6) Finally, the authors did not mention whether some LRAs, such as immune checkpoint inhibitors (anti-PD-1, anti-CTLA-4) or P-TEFb release agents (JQ1, I-BET…), have been shown to act differently on distinct HIV-1 subtypes.
Author Response
We will like to thank the constructive criticism of the reviewer. We believe that we have addressed all the reviewers concern and that our revised manuscript will be now suitable for publication.
In line 102, the authors stated: “Subtype C and subtype A rarely undergo this switch; subtype C viruses favor CCR5 more extremely throughout infection than subtype A”. From lines 103 to 106, the authors explained the V3 loop variation in subtype C, however they did not explain it for subtypeWe have added more information explaining the V3 loop variation for subtype A.
The authors should mention the major LTR differences found within different HIV-1 subtypes such as the ones mentioned in their cited references 44 (STAT-binding sites), 45 (numbers of NF-kappaB sites), 47 (competition between NF-kappaB binding and Tat transactivation capacity), 49 (promoter architecture) and 50 (response to reactivation from latency).We thank the reviewer for this comment and have made the changes suggested. We feel that it has helped to improve the clarity of the manuscript by providing both explanations of the major differences in LTR, in addition to a visual representation of the LTR (Figure 2).
The authors could add some information about HIV-1 integration site specificity and the efficiency of the viral immune response depending on the viral subtype.
We have added a section describing integration site and the literature about immune responses generated depending on viral subtype, and feel that this has enhanced the overall manuscript.
Line 342: remove capital letter from “for a review”.
We have made this change.
A figure or a table illustrating the review should be added (such as a scheme representing the LTR architecture of the different subtypes with the differences in transcription factor binding sites).
We have added a graphical representation of the LTR (Figure 2) to help illustrate the differences in LTR architecture as well as four tables to summarize the different studies.
Point 6) Finally, the authors did not mention whether some LRAs, such as immune checkpoint inhibitors (anti-PD-1, anti-CTLA-4) or P-TEFb release agents (JQ1, I-BET…), have been shown to act differently on distinct HIV-1 subtypes.
We have added this information in the LRAs section and thank the reviewer for the suggestion.
Reviewer 2 Report
Sarabia and Bosque have written a comprehensive overview on the subject of the impact of HIV-1 subtype diversity on HIV cure strategies. This is of high importance because subtype B is most commonly investigated yet comprises only up to 10% of worldwide infections and it is currently not clear if this impacts global cure strategies.
Some minor edits are suggested by this reviewer:
Page 2, line 93: It would be appropriate to include a reference for the co-receptor switch
Page 3, line 103: period is missing at the end of the sentence “…infection than subtype A [15, 22, 39]”
Page 3, line 116: typo, “van Opinjen” should be “van Opijnen”
Page 5, lines 226-227: It would be appropriate to include a reference for the definition of a latent HIV reservoir.
Author Response
We will like to thank the constructive criticism of the reviewer. We believe that we have addressed all the reviewers concern and that our revised manuscript will be now suitable for publication.
2, line 93: It would be appropriate to include a reference for the co-receptor switch
We have made this change.
2.Page 3, line 103: period is missing at the end of the sentence “…infection than subtype A [15, 22, 39]”
We have fixed this typo.
3.Page 3, line 116: typo, “van Opinjen” should be “van Opijnen”
We have fixed this typo and apologize for any confusion.
Page 5, lines 226-227: It would be appropriate to include a reference for the definition of a latent HIV reservoir.We have made this change.
Reviewer 3 Report
This review briefly describe HIV-1 latency and latency reversal. It focuses also on the subtype-specific factors which may affect viral latency establishment and subsequent latency reactivation.
This reviewer feel that in the current state, this manuscript is not suitable for publication in the viruses, and need through revision.
Major suggestions:
Authors should explain the referred studies with greater clarity. Authors are unable to provide figures and tables to support their statements. Lack of figures and tables is a major drawback of this manuscript. Line numbers 110 to 169 and 275 to 400 could have been summarized in table or figure format to provide clarity to the readers. There are lots of statements throughout the manuscript that are not supported by the literature reviewed, which is matter of concern. For the sake of radar’s understanding, authors should provide their conclusions based on literature available, instead of just stating the results and observations of researchers. Certain findings are mentioned that were neither extended nor confirmed by other investigators, especially in “the viral long Terminal (LTR)” section. However, this tendency exists throughout the review, which makes very difficult to understand and conclude. LTR section in particular require thorough revision. There are many typos and grammatical issues throughout the manuscript, they need to be addressed. Need full forms of numerous acronym and abbreviations, such as SMAC, IAP, BIR, BIRC2, STAT etc. The title of the manuscript is irrelevant to the content of the manuscript, as authors has mentioned a number of studies, which annulled the role of HIV-1 subtypes in both establishing and reversing HIV-1 latency. Additionally, in a recent study Bachmann N et.al 2019 (Nat. Communication) investigated the influence of HIV-1 subtype on the HIV-1 latent reservoir long-term dynamics, but found no significant correlation. Moreover, their conclusions contradicts the hypothesis authors proposed based on their literature search. A number of sentences are not clear, such as “The prevalence of these variants was not uniformly distributed among subtypes. These subtle changes in sequence resulted in re-targeting from gene-rich areas to gene-poor regions, characterized by heterochromatin and H3K9 bi and trimethylation. Interestingly, these variants were associated with faster disease progression in a chronic infection subtype C cohort [59]“Activation of PKC induces activation of IKK which causes degradation of IkBα, allowing NF-kB to be released, phosphorylated, and translocated to the nucleus. In the setting of HIV-1 infection, NF-kB can be recruited to NF-B binding sites present in the viral LTR, thus appropriating NF-kB to promote its own transcription”.
Additionally, the following statement is highly immature and an isolated incident, as there is no sufficient premise to support “Since the prevalence of these integrase variants differs among subtypes (the global re-targeting variants INS119G occur most in subtype B, and INR231G in subtype C [59]), this could affect integration site patterns from subtype to subtype. Differences in integration site selection could likely impact latency reversal efforts”. The following statement need further clarification “Interestingly, subtype C was least effective at overcoming APOBEC3G mediated viral inhibition compared to subtype B or subtype AG in this system. This sensitivity to APOBEC3G may affect the intactness of the integrated subtype C viruses, thus potentially limiting inducibility of the latent provirus”. The flow of manuscript is highly inconsistent, For examples “HIV-1 Viral Vpr is involved in transport of the pre-integration complex to the nucleus , and cell cycle arrest [60]. Subtype C Vpr exhibited significant sequence changes from subtype B in a study by Bano et al [63]. They examined the ability of subtype C-Vpr and subtype-B Vpr to transactivate the subtype-specific HIV-1 LTR. Using a luciferase reporter LTR, they found that subtype B more potently transactivated the HIV-1 LTR”.
“Finally, it has been shown that the presence of accessory genes may influence the establishment of latent infections in a primary cell model that uses fetal thymocytes [74]. The genetic sequence of accessory proteins can vary from 5% to 35% by subtype, potentially affecting function [75, 76]. Thus, we speculate that perhaps non-B viral proteins may have varying degrees of impact on latency. Further studies using non-subtype B viruses are needed to address this possibility”.
They should refer all the relevant studies, otherwise their conclusions considered biased.Author Response
We will like to thank the constructive criticism of the reviewer. We believe that we have addressed all the reviewers concern and that our revised manuscript will be now suitable for publication.
Reviewer 3
Authors should explain the referred studies with greater clarity.
We have made significant changes to the manuscript to better explain the referred studies.
Authors are unable to provide figures and tables to support their statements. Lack of figures and tables is a major drawback of this manuscript. Line numbers 110 to 169 and 275 to 400 could have been summarized in table or figure format to provide clarity to the readers.We thank the reviewer for the kind suggestion and have added several figures and tables that we feel have enhanced the quality of the manuscript, providing further clarity to the readers.
There are lots of statements throughout the manuscript that are not supported by the literature reviewed, which is matter of concern.We have thoroughly reviewed the statements made in the manuscript to address this concern.
For the sake of radar’s understanding, authors should provide their conclusions based on literature available, instead of just stating the results and observations of researchers.We have thoroughly reviewed the statements made in the manuscript to address this comment.
Certain findings are mentioned that were neither extended nor confirmed by other investigators, especially in “the viral long Terminal (LTR)” section. However, this tendency exists throughout the review, which makes very difficult to understand and conclude.We agree with the reviewer in that more research needs to be done beyond what currently exists in the literature. The review at present is aimed to draw attention to this and summarize the state of the field with regard to subtype and latency.
LTR section in particular require thorough revision. There are many typos and grammatical issues throughout the manuscript, they need to be addressed.We have reviewed the entire manuscript to correct typos, flow of the manuscript, and grammatical issues and apologize for any confusion.
Need full forms of numerous acronym and abbreviations, such as SMAC, IAP, BIR, BIRC2, STAT etc.We thank the reviewer for bringing this to our attention and have made this change.
The title of the manuscript is irrelevant to the content of the manuscript, as authors has mentioned a number of studies, which annulled the role of HIV-1 subtypes in both establishing and reversing HIV-1 latency.We feel that the title is germane to the literature reviewed in the manuscript, and the studies cited do not eliminate the possibility that subtype influences the establishment of HIV latency as well as its reversing.
Additionally, in a recent study Bachmann N et.al 2019 (Nat. Communication) investigated the influence of HIV-1 subtype on the HIV-1 latent reservoir long-term dynamics but found no significant correlation. Moreover, their conclusions contradicts the hypothesis authors proposed based on their literature search.We thank the reviewer for bringing this interesting study to our attention, we have included it in this review. The results from this paper show that there is no significant association between reservoir size and subtype, but there is a trend towards smaller reservoirs in non-B subtypes. Additionally, the rate of decay was faster in non-B subtypes (even after ethnicity was controlled for, in a multivariable analysis), suggesting another mechanism that may be subtype-specific for this viral reservoir decay rate. We feel the inclusion of this new study greatly enhances our review.
A number of sentences are not clear, such as “The prevalence of these variants was not uniformly distributed among subtypes. These subtle changes in sequence resulted in re-targeting from gene-rich areas to gene-poor regions, characterized by heterochromatin and H3K9 bi and trimethylation. Interestingly, these variants were associated with faster disease progression in a chronic infection subtype C cohort [59]“Activation of PKC induces activation of IKK which causes degradation of IkBα, allowing NF-kB to be released, phosphorylated, and translocated to the nucleus. In the setting of HIV-1 infection, NF-kB can be recruited to NF-B binding sites present in the viral LTR, thus appropriating NF-kB to promote its own transcription”.We have made changes to these sentences to improve the clarity of the manuscript.
Additionally, the following statement is highly immature and an isolated incident, as there is no sufficient premise to support “Since the prevalence of these integrase variants differs among subtypes (the global re-targeting variants INS119G occur most in subtype B, and INR231G in subtype C [59]), this could affect integration site patterns from subtype to subtype. Differences in integration site selection could likely impact latency reversal efforts”.Respectfully, the literature with respect to whether there are subtype-specific functional differences in integrase is limited. The statement is speculation, as this avenue of research is fairly new. We thank the reviewer for their professional and constructive criticism, we have made changes to the statements to better explain the study.
The following statement need further clarification “Interestingly, subtype C was least effective at overcoming APOBEC3G mediated viral inhibition compared to subtype B or subtype AG in this system. This sensitivity to APOBEC3G may affect the intactness of the integrated subtype C viruses, thus potentially limiting inducibility of the latent provirus”.We have made changes to these sentences to improve the clarity of the manuscript.
The flow of manuscript is highly inconsistent, For examples “HIV-1 Viral Vpr is involved in transport of the pre-integration complex to the nucleus , and cell cycle arrest [60]. Subtype C Vpr exhibited significant sequence changes from subtype B in a study by Bano et al [63]. They examined the ability of subtype C-Vpr and subtype-B Vpr to transactivate the subtype-specific HIV-1 LTR. Using a luciferase reporter LTR, they found that subtype B more potently transactivated the HIV-1 LTR”.“Finally, it has been shown that the presence of accessory genes may influence the establishment of latent infections in a primary cell model that uses fetal thymocytes [74]. The genetic sequence of accessory proteins can vary from 5% to 35% by subtype, potentially affecting function [75, 76]. Thus, we speculate that perhaps non-B viral proteins may have varying degrees of impact on latency. Further studies using non-subtype B viruses are needed to address this possibility”.
We have made changes to the manuscript to improve the overall flow.
They should refer all the relevant studies, otherwise their conclusions considered biased.
We have included all relevant studies to the best of our ability and leave the conclusions up to the reader based on the literature presented. This review is aimed at shining a light on what work has been done and what needs to be further explored, in the efforts of developing globally relevant cure strategies.
Round 2
Reviewer 3 Report
The revised version sounds adequate to be published in this esteemed journal.